# Analysis of Multi-Mycotoxins in Commonly Consumed Spices Using the LC-MS/MS Method for Assessing Food Safety Risks

**DOI:** 10.3390/microorganisms11071786

**Published:** 2023-07-11

**Authors:** Burak Demirhan, Buket Er Demirhan

**Affiliations:** Department of Pharmaceutical Basic Sciences, Faculty of Pharmacy, Gazi University, Ankara 06330, Turkey; erbuket@gazi.edu.tr

**Keywords:** food safety, LC-MS/MS, mycotoxin, spice

## Abstract

Mycotoxins are secondary metabolites produced by certain fungal species. In this study, the aim was to investigate mycotoxins, which pose a serious health problem. For this purpose, a total of 140 spice samples (black pepper, red pepper, cumin, and turmeric) purchased from Ankara, Turkey, were analyzed for specific mycotoxins (aflatoxin B1-AFB1, aflatoxin B2-AFB2, aflatoxin G1-AFG1, aflatoxin G2-AFG2, ochratoxin A-OTA, zearalenone-ZEN) using an LC-MS/MS multi-mycotoxin method. The *Staphylococcus* spp. and *Micrococcus* spp. counts in the spice samples were also analyzed using the conventional culture method. The contamination levels of AFB1 ranged from not detected (ND) to 39.12 μg/kg; AFB2 ranged from ND to 2.10 μg/kg; AFG1 ranged from ND to 0.92 μg/kg; AFG2 ranged from ND to 3.67 μg/kg; OTA ranged from ND to 39.79 μg/kg; ZEN ranged from ND to 11.16 μg/kg. The maximum residue limit for AFB1 (5 μg/kg) determined according to the Turkish Food Codex (TFC) was exceeded in five samples of red pepper, two samples of black pepper, and one sample of turmeric. Furthermore, it was determined that three samples of red pepper and one sample of black pepper exceeded the maximum limits for total aflatoxin (10 μg/kg) and OTA (15 μg/kg) specified in the TFC.

## 1. Introduction

Spices and condiments represent a crucial aspect of human nutrition and are endowed with a wealth of phytochemical constituents [1]. The employment of spices in culinary practices for enhancing flavors, colors, and aromas can be traced back to time immemorial [2,3]. Furthermore, they serve as preservatives, medicinal agents, and cosmetic ingredients [4,5]. From a global trade perspective, black pepper, paprika, cumin, turmeric, nutmeg, cinnamon, ginger, cloves, and coriander emerge as the most economically significant spice crops [1].

Spices are subjected to attacks by diverse microorganisms, which can be attributed to deficient production processes, inadequate storage conditions, and insufficient drying times [6]. Multiple factors, such as unsanitary equipment, inadequate hygiene conditions, substandard storage conditions, and suboptimal handling practices, are recognized as potent sources of spice contamination [7]. The collection, processing, and marketing phases of spice production result in significant exposure to environmental microbial contaminants such as dust, wastewater, and animal as well as human excrement [8]. The level of microbial contamination can vary depending on factors such as geographic location, the year of manufacture, pre-drying harvest, and storage conditions [9]. Nonetheless, some spices provide an optimal environment for the development of mold fungi and their subsequent growth [3]. Pre-harvest, post-harvest, drying, processing, packaging, storage, and the transportation stages can all serve as potential avenues for mold contamination [6].

Mycotoxins are secondary metabolites produced by certain fungal species known for their toxic, carcinogenic, mutagenic, and teratogenic effects [10,11]. These harmful substances can be found in various food products, such as baby foods, peanut butter, sesame samples, nut-based foods, red pepper products, milk, and cheese [11,12,13,14,15,16]. Among mycotoxins, aflatoxins (AFs) and ochratoxin A (OTA) are commonly associated with spices [17,18]. AFs, which are predominantly produced by Aspergillus species including *Aspergillus flavus*, *A. parasiticus*, and occasionally *A. nomius*, are recognized as the most toxic group of mycotoxins [19]. OTA, on the other hand, is a significant mycotoxin from a public health perspective known for its nephrotoxic properties and its ability to cause kidney damage [20,21]. Apart from AFB1, AFG1, OTA, and ZEN are frequently detected mycotoxins in spices [22]. The mycotoxin ZEN, which is produced by Fusarium species in food and reported to have estrogenic, genotoxic, mutagenic, teratogenic, immunosuppressive, and hematotoxic effects, is classified as Group 3 (not classifiable according to its carcinogenicity to humans) by the International Agency for Research on Cancer (IARC) [21]. It is important to note that these mycotoxins have been identified as endocrine disruptors in humans [23].

It is widely recognized that mycotoxins pose significant threats to human and animal health, as well as having notable economic implications [24]. Consequently, many countries around the world have established regulatory limits for mycotoxins in food. In line with the European Union Commission Regulation on Determination of the Maximum Limits of Certain Contaminants in Foods (numbered 1881/2006/EC), the Turkish Food Codex (TFC) Contaminants Regulation has been developed [25]. The TFC sets maximum limits for mycotoxins in various spices, such as black pepper (*Piper* spp.), red pepper (*Capsicum* spp.), nutmeg (*Myristica fragrans*), ginger (*Zingiber officinale*), turmeric (*Curcuma longa*), and spice mixtures containing these ingredients. The prescribed limits are 10 µg/kg for total aflatoxins (B1 + B2 + G1 + G2) and 5 µg/kg for AFB1 [26]. Additionally, the TFC specifies a maximum limit of 15 µg/kg for OTA in the aforementioned spices or mixtures containing them.

The primary aim of this investigation was to examine the potential presence of hazardous mycotoxins, including AFs, OTA, and ZEN, as well as the contamination of *Staphylococcus* spp. and *Micrococcus* spp., in black pepper, red pepper, cumin, and turmeric available in the Turkish market. Furthermore, it also aimed to evaluate the exposure occurring in adults through the consumption of these spices, in order to determine compliance with the maximum permissible levels of mycotoxins set by the Turkish Food Codex and assess the associated potential risks.

## 2. Materials and Methods

### 2.1. Samples

A total of one hundred and forty packaged spice samples were collected between April and July 2021 from different local markets in Ankara, Turkey. The samples consisted of 40 samples of red pepper (10 from each of the four brands, labeled A, B, C, and D), 40 samples of black pepper (likewise from brands A–D), 40 samples of cumin (also from brands A–D), and 20 samples of turmeric (10 from each of two brands, labeled A and B). These brands are all major manufacturers whose products are sold throughout Turkey, and each sample had a unique serial number. All samples were stored in their original packaging in a cool, dry, and dark place at room temperature until analysis.

### 2.2. Sample Preparation

The LC-MS/MS Jasem analysis kit (SEM, Istanbul, Turkey) instructions were employed to prepare spice samples for analysis [27]. For the detection of 6 mycotoxins (AFB1, AFB2, AFG1, AFG2, OTA, and ZEN) in black pepper, red pepper, and cumin, as well as AFs (AFB1, AFB2, AFG1, and AFG2) and ZEN in turmeric, 5.0 g of homogenized spice sample was transferred to a 50 mL centrifuge tube. Twenty milliliters of reagent 1 (JSM FO 9704; SEM, Istanbul, Turkey) was added to the sample, followed by shaking for 15 min using a multi-shaker. After shaking, the mixture was centrifuged at 3000 rpm for 5 min at room temperature. The clear supernatant obtained via centrifugation was filtered through a 0.45 µm nylon membrane filter and directly injected into the LC-MS/MS system. To determine OTA mycotoxin contamination levels in turmeric samples, 200 mg of each homogenized sample was diluted with 4 mL of distilled water and mixed well for 2 min. Then, 6 mL of tetrabutylmethylether (TBME) was added and mixed well for 15 min, followed by centrifugation for 5 min at 4000 rpm. An amount of 1 mL of the aqueous phase was taken, and 1 mL of reagent 1 (JSM-FO-9704; SEM, Istanbul, Turkey) was added to it. The mixture was filtered through a 0.45 μm nylon membrane filter before being injected into the LC-MS/MS system.

### 2.3. LC-MS/MS Analysis

To assess the presence of mycotoxins in spice samples, we utilized a commercially available multi-mycotoxin LC-MS/MS Jasem kit (SEM, Istanbul, Turkey) capable of detecting six mycotoxins: AFB1, AFB2, AFG1, AFG2, OTA, and ZEN [27]. The LC-MS/MS system consisted of an Agilent 1290 Infinity UHPLC coupled with an Agilent 6460 triple-quadrupole mass spectrometer with electrospray ionization (Agilent Technologies, Santa Clara, CA, USA). The chromatographic separation was performed on a Jasem multi-mycotoxin column (JSM-FO-9775; SEM, Istanbul, Turkey) at a flow rate of 0.5 mL/min. The injection volume for each sample was 10 μL, and the column temperature was maintained at 35 °C. The quantitative data analysis was carried out using MassHunter 6470 software version 10.1 (Agilent Technologies, Santa Clara, CA, USA).

UHPLC gradient conditions were set as follows: 20% B from 0 to 1 min, 2–95% B from 1 to 4 min, 95% B from 4 to 7 min, and 20% B from 7 to 7.1 min, followed by a hold at 20% B from 7.1 to 12 min to optimize analyte separation (AFB1, AFB2, AFG1, AFG2, ZEN, OTA). The ion source parameters for analysis in the mass spectrometer were set to +3500 V and −3500 V for positive and negative capillary voltage, 40 psi for nebulizer pressure, 300 °C and 11 L/min for drying gas temperature and flow rate, and 400 °C and 11 L/min for sheath gas temperature and flow rate. LC-MS/MS chromatograms for the six mycotoxins are presented in Figure 1. The validation parameters assessed for the spice samples included linear range, the coefficient of determination (R^2^), relative standard deviation (RSD), recovery (Rec), limit of detection (LOD), and limit of quantification (LOQ), as shown in Table 1. Recovery was calculated from a calibration concentration of 0.125 ppb for AFB1, AFB2, AFG1, AFG2, and OTA, and a calibration concentration of 2.5 ppb for ZEN. The standard deviation of 10 repeated injections was determined, and the % RSD was calculated. 

### 2.4. Microbiota Tests

All of the spice samples were transported to the laboratory to conduct microbiological investigations. *Staphylococcus* spp. and *Micrococcus* spp. counts were determined using traditional methods. The detection of *Staphylococcus* spp. and *Micrococcus* spp. was performed using Baird-Parker agar medium (Merck 105406, Darmstadt, Germany) supplemented with egg yolk tellurite (Merck 103785, Darmstadt, Germany). Ten-gram portions of spice samples were aseptically weighed and mixed with 90 mL of sterile Maximum Recovery Diluent (MRD, Merck 112535, Darmstadt, Germany). The sample was diluted in a ratio of 1:10 with the MRD and homogenized using a vortex mixer. After incubating the samples for 48 h at 37 °C, the colony counts were calculated. For *Staphylococcus* spp., characteristic black colonies on the Baird-Parker agar plate surrounded by a clear zone were selected, while for *Micrococcus* spp., black colonies without a clear zone were selected [28]. The colonies of bacteria were counted and expressed as log10 CFU/g of sample.

### 2.5. Dietary Exposure Assessment

Based on the available consumption data, the estimated mycotoxin exposure through the consumption of red pepper, black pepper, cumin, and turmeric spices was determined [29]. The estimation of mycotoxin intake through diet was calculated using the following formula:



$\begin{array}{cc} Estimated\;daily & intake\;\left(\frac{ng}{kg}\right)\\ & =\frac{\mathrm{mycotoxin}\;\mathrm{level}\;\mathrm{in}\;\mathrm{food}\left(\frac{\mathrm{ng}}{g}\right)\;\times \;\mathrm{food}\;\mathrm{consumption}\left(\frac{\frac{g}{\mathrm{person}}}{\mathrm{day}}\right)}{Average\;body\;weight\;\left(kg\right)} \end{array}$



According to the data from the Turkish Statistical Institute on average weight by age group and gender for individuals aged 15–75 years, the average body weight was taken as 73.5 kg [30]. The daily spice consumption was assumed to be 10 g [4].

### 2.6. Margin of Exposure

The carcinogenic and toxic effects of mycotoxins were determined based on the margin of exposure (MOE) assessment. This value was calculated by dividing the benchmark lower dose limit (BMDL) established for mycotoxins by the probable daily intake value [31]. When the MOE value is equal to or greater than 10,000, it implies a lower health concern, whereas a value below 10,000 indicates significant health risks associated with exposure to that contaminant [32]. 

### 2.7. Statistical Analysis

The SPSS program was used in the statistical evaluation of the data obtained as a result of the analysis. An independent sample *t*-test and a one-way ANOVA test were used to statistically evaluate the difference between the groups [33].

## 3. Results

The analysis of six important mycotoxins, namely AFB1, AFB2, AFG1, AFG2, OTA, and ZEN, was performed using the LC-MS/MS method, while *Staphylococcus* spp./*Micrococcus* spp. analysis was conducted using the classical culture method. A total of 140 packaged spice samples, including red pepper, black pepper, cumin, and turmeric, were collected from local markets in Ankara. The mycotoxin distribution of the spice samples is presented in Table 2.

The mean levels of AFB1 in the spice samples ranged from not detected (ND) to 6.03 ± 4.18 µg/kg. Red pepper samples showed the highest mean AFB1 levels (6.01 ± 2.99 µg/kg and 6.03 ± 4.18 µg/kg), whereas all red pepper brands had a mean AFB1 value of 3.76 ± 1.27 µg/kg. The difference between the AFB1 mean levels of brands A and D in red pepper samples was statistically significant (*p* < 0.05). Similarly, the difference between the AFB1 mean levels of brands B and C in black pepper was statistically significant (*p* < 0.05), and brand B had the highest mean AFB1 value in black pepper, cumin and turmeric samples.

AFB2 mean values in all spice samples ranged from ND to 0.75 ± 0.34 µg/kg. All of the red pepper sample brands had AFB2 mean values (0.28 ± 0.02 µg/kg, 0.55 ± 0.13 µg/kg, 0.75 ± 0.34 µg/kg, and 0.34 ± 0.06 µg/kg), and the highest AFB2 mean level was detected in brand C (0.75 ± 0.34 µg/kg). In the black pepper samples, the mean AFB2 value of brand A was 0.68 ± 0.14 µg/kg, and 0.16 µg/kg AFB2 was determined in one sample in brand B. AFB2 and AFG1 could not be detected in cumin and turmeric samples. In black pepper samples, the mean AFG1 value of brand C was 0.63 ± 0.29 µg/kg.

The mean levels of AFG1 in red pepper samples were observed to range between 0.15 ± 0.01 µg/kg and 0.30 ± 0.04 µg/kg. Brand A displayed the lowest AFG1 mean value among the red pepper brands, and the statistical analysis indicated a significant difference (*p* < 0.05) between brand A and brands B and C.

AFG2 was not detected in any of the cumin samples. The mean levels of AFG2 in red pepper, black pepper, and turmeric samples ranged from ND to 1.52 ± 0.80 µg/kg. The highest mean level of AFG2 was observed in the black pepper brand C, while brand B had the highest mean levels of AFG2 in red pepper samples.

The mean levels of OTA ranged from ND to 11.68 ± 4.42 µg/kg. The highest OTA mean levels were found in brands B and D of red pepper samples, as well as in brand B of black pepper and cumin samples. ZEN was detected at 6.20 ± 1.42 µg/kg only in brand A of turmeric samples.

Table 3 shows the distribution of *Staphylococcus* spp. and *Micrococcus* spp. in the spice samples. *Staphylococcus* spp. growth was observed only in the black pepper samples of brands B and D. The mean value of *Staphylococcus* spp. in brand B was higher than that in brand D, and the difference between the two brands was statistically significant (*p* < 0.05). The mean values of *Micrococcus* spp. were determined in the black pepper samples of brands A, B, and D. The difference between the brands was statistically significant (*p* < 0.05), and brand B had the highest *Micrococcus* spp. value.

In red pepper samples, there was a statistically significant difference (*p* < 0.05) between brands B, C, and D in terms of *Staphylococcus* spp. levels. Moreover, all brands showed a statistically significant difference (*p* < 0.05) in terms of *Micrococcus* spp. levels. Notably, brand B exhibited the highest mean levels of both *Staphylococcus* spp. and *Micrococcus* spp. in red pepper samples.

The difference in terms of *Staphylococcus* spp. among brands B, C, and D of cumin samples was statistically significant (*p* < 0.05). Similarly, the difference in terms of *Micrococcus* spp. among these brands was statistically significant (*p* < 0.05). Brand B exhibited the highest mean levels of both *Staphylococcus* spp. and *Micrococcus* spp. in cumin samples. In turmeric samples, growth of *Staphylococcus* spp. and *Micrococcus* spp. was detected. Brand B had a higher microbiological value compared to brand A in turmeric samples, although the difference between the two brands was not statistically significant (*p* > 0.05).

The estimated daily intake levels of analyzed mycotoxins in spice samples are provided in Table 4. When evaluated for AFB1, red pepper spice showed the highest EDI level, while black pepper spice exhibited the lowest EDI value. AFB2 and AFG1 were detected only in black pepper and red pepper samples, with black pepper having a higher EDI value. AFG2 was detected in all spice samples except cumin, with black pepper having the highest value and turmeric having the lowest. The EDI value of OTA mycotoxin was found in all spice samples except turmeric, with red pepper samples showing the highest value. On the other hand, ZEN mycotoxins were exclusively found in the turmeric spice.

MOE is a tool used to assess potential safety concerns arising from the presence of chemical substances in food and feed. In 2005, the European Food Safety Authority (EFSA) stated that an MOE equal to or greater than 10,000 for both genotoxic and carcinogenic substances signifies low concern for public health and may be considered as a low priority for risk management measures [32]. The margin of exposure values for spice samples are provided in Table 5. When examining the margin of exposure values of mycotoxins in spices, it was found that the OTA MOE value in black pepper and cumin was above 10,000, indicating a low risk. The highest risk ratio was observed for AFB1 in red peppers, followed by AFB1 in turmeric, AFB1 and total aflatoxins in cumin, AFB1 in black pepper, total aflatoxins in red pepper and black pepper, and total aflatoxins in turmeric. The OTA risk in red peppers was also found to be close to 10,000. 

## 4. Discussion

Spices hold a pivotal position in agricultural economies, encompassing esteemed varieties such as black pepper, red pepper, nutmeg, cumin, cinnamon, ginger, turmeric, cloves, and coriander, which collectively grace the global spice landscape. Given their significant economic worth, spices are unfortunately susceptible to pervasive adulteration practices. Tropical regions in particular witness elevated contamination levels in spice production. This contamination may stem from various sources, including cross-contamination with dust, wastewater, and exposure to physical, biological, and chemical hazards like avian species, rodents, and insects. To safeguard against such adulterations and ensure quality production conditions, countries have devised a multitude of rigorous quality standards and regulations. Within this context, the deployment of diverse detection methodologies assumes paramount importance in unearthing instances of adulteration in spices and spice-based products [1].

There are numerous studies available in the international literature addressing mycotoxin contamination of spices, and several noteworthy studies stand out (Table 6).

In our current investigation, AFB2 concentrations were measured as 0.68 ± 0.14 µg/kg in brand A and 0.16 µg/kg in brand B black pepper samples. Moreover, the mean OTA levels in black pepper samples were determined to be 7.89 ± 0.97 µg/kg for brand B and 5.46 ± 2.83 µg/kg for brand D. Notably, the OTA and AFB2 values obtained in our study surpass those reported by Pantano et al. [34]. 

Pickova et al. [35] conducted an analysis of OTA in spices in the Czech Republic. They reported that 31% of the samples, consisting of 19 different spice types, contained OTA in the range of 0.11–38.46 ng/g. The mean levels of OTA in turmeric, cumin, and black pepper samples were reported as 19.82 ng/g, 0.46 ng/g, and 0.31 ng/g, respectively. In our study, the OTA levels in turmeric, cumin, and black pepper samples were found to be 0.49 µg/kg (in one sample), 3.19 ± 0.72 µg/kg, and 6.27 ± 1.89 µg/kg, respectively. Compared with the findings of Pickova et al. [35], it is evident that the contamination of turmeric samples with OTA is low, while the contamination of cumin and black pepper samples is high. 

It was observed that the aflatoxin and ZEN values in our study are lower compared to those reported by El Darra et al. [36], while the OTA values are similar. 

Nguegwouo et al. [37] conducted a study on the contamination of black pepper and white pepper samples with OTA, reporting that 10% of black pepper samples were contaminated with OTA at levels ranging from 1.15 to 1.91 μg/kg. In comparison, our study indicates a higher level of OTA contamination in black pepper samples than that found by Nguegwouo et al. [37]. Our study reveals that the OTA contamination of black pepper and red pepper samples reported by Jalili [38] is lower, while the contamination of turmeric is higher. According to the findings obtained by Barani et al. [38], it can be observed that the levels of AFB1 and AFB2 in red peppers are high compared to our study, while the levels of AFB1 and AFB2 in black peppers are low.

Jalili and Jinap [40] conducted a study to investigate the natural occurrence of aflatoxins and ochratoxin A in 80 commercial dried peppers. Upon comparison with the results of our study, it is evident that the average levels of AFB1, AFB2, and OTA are similar, while our AFG1 value is lower and our AFG2 value is higher.

Upon comparison with our study, it is evident that the aflatoxin and OTA values in our red peppers were generally lower than those reported by Ozbey and Kabak [45]. It is notable that our black pepper and cumin samples exhibited relatively elevated levels of mycotoxins.

Yentur et al. [14] conducted a study to investigate AFB1 levels in 190 red pepper products. They reported that one sample of pepper sauce exceeded 5 ppb.

In another study, Kurşun and Mutlu [45] investigated 134 spice samples to assess total aflatoxin contamination. In our study, the total aflatoxin contamination levels in red pepper and cumin samples were lower than those reported by Kurşun and Mutlu [45], while the contamination levels in black pepper samples were generally comparable.

Çolak et al. [46] analyzed a total of 84 spice samples using ELISA and HPLC methods to determine the levels of total aflatoxin and AFB1. They reported that red pepper samples had total aflatoxin levels ranging from 0.3 to 44.7 µg/kg and 0.7 to 46.8 µg/kg according to HPLC and ELISA methods, respectively. For black pepper samples, they found total aflatoxin levels between 0.1 and 15.3 µg/kg and 0.3 and 16.7 µg/kg through HPLC and ELISA methods, respectively. 

Microorganisms commonly isolated from spices include *Salmonella* spp., *Escherichia coli*, *Aspergillus niger*, *Rhizopus* spp., *Clostridium perfringens*, *Shigella dysenteriae*, *Staphylococcus aureus*, thermotolerant coliforms, *Bacillus cereus*, *Bacillus coagulans*, *Bacillus polymyxa*, and *Bacillus subtilis*. Inadequate storage conditions contribute to the development and proliferation of these microorganisms, leading to the production of toxins in food. Given that spices can be susceptible to microbial contamination due to inappropriate conditions and unhygienic practices during processing and transportation, meticulous monitoring of these stages is of the utmost importance to safeguard both product quality and public health [1].

Demir et al. [47] conducted an investigation to assess the microbiological quality of 120 spice samples obtained from open markets. They reported mean levels of *Staphylococcus*/*Micrococcus* to be 5.52 log CFU/g, 4.01 log CFU/g, and 4.89 log CFU/g in black pepper, red chili pepper, and cumin samples, respectively. A comparison of their findings with our study reveals higher levels in the results reported by Demir et al. [47]. It is worth noting that the spices used in our study were commercially packaged products, which may explain the relatively lower contamination levels observed compared to those reported by Demir et al. [47].

Beki and Ulukanli [48] conducted a comprehensive assessment of microorganism counts and the presence of certain pathogens in spices sold in open markets in Kars. They reported that the contamination levels of *Staphylococcus*/*Micrococcus* ranged from 10^2^ CFU/g to 10^7^ CFU/g in red pepper samples, from 10^3^ CFU/g to 10^7^ CFU/g in black pepper samples, and from 10^2^ CFU/g to 10^6^ CFU/g in other spice samples. Similarly, Kara et al. [49] examined the microbiological quality of 250 spices and found *Staphylococcus*/*Micrococcus* contamination levels to be 2.03 log CFU/g in red pepper flakes, 2.73 log CFU/g in cumin samples, and 1.11 log CFU/g in black pepper samples. In comparison to the findings of our study, it is evident that the contamination levels in the spice samples analyzed by Kara et al. [49] were relatively lower. 

Sospedra et al. [50] reported that they isolated *Staphylococcus aureus* in 2 out of 30 spice samples collected from Spanish markets, highlighting the potential health risks associated with the presence of pathogenic microorganisms like *S. aureus* in spices.

## 5. Conclusions

The detection of mycotoxins in food is a matter of great importance due to their inherent toxicity. These unavoidable contaminants cannot be completely eliminated. Therefore, the implementation of effective monitoring programs and stringent legal regulations can significantly contribute to the reduction in mycotoxin levels in food.

Comparison of the results with the maximum limits specified in the Turkish Food Codex and European Union Regulations revealed that some brands exceeded the mycotoxin parameters. To ensure food safety, it is recommended strict hygiene practices and quality control measures be maintained during the production of spices in Turkey.

In the present study, a total of 140 samples of spices from various brands were collected in Ankara, Turkey, and analyzed for the presence of mycotoxins, including AFB1, AFB2, AFG1, AFG2, OTA, and ZEN. Upon evaluating the mycotoxin results, it was observed that brand B consistently exhibited higher levels compared to other brands. This could be attributed to factors such as raw material contamination and inadequate hygienic conditions during production. Furthermore, an analysis based on spice type revealed that red pepper samples showed a greater incidence of mycotoxin contamination. It is well documented that unfavorable environmental conditions during the production of red peppers, such as moisture and temperature, significantly contribute to the growth of mycotoxin-producing molds. As a result, red peppers are considered to be of critical importance in terms of public health. Additionally, in terms of microbiological analysis, brand B generally exhibited higher mean values of *Staphylococcus* spp./*Micrococcus* spp.

The obtained results were subjected to a meticulous comparison with the stringent maximum limits defined in both the Turkish Food Codex and the Regulations of the European Union, and it was observed that certain brands exhibited mycotoxin parameters surpassing these established thresholds. Consequently, it is highly recommended impeccable hygiene practices and quality standards be rigorously upheld and prioritized throughout all stages of spice production within our country.

## Figures and Tables

**Figure 1 microorganisms-11-01786-f001:**
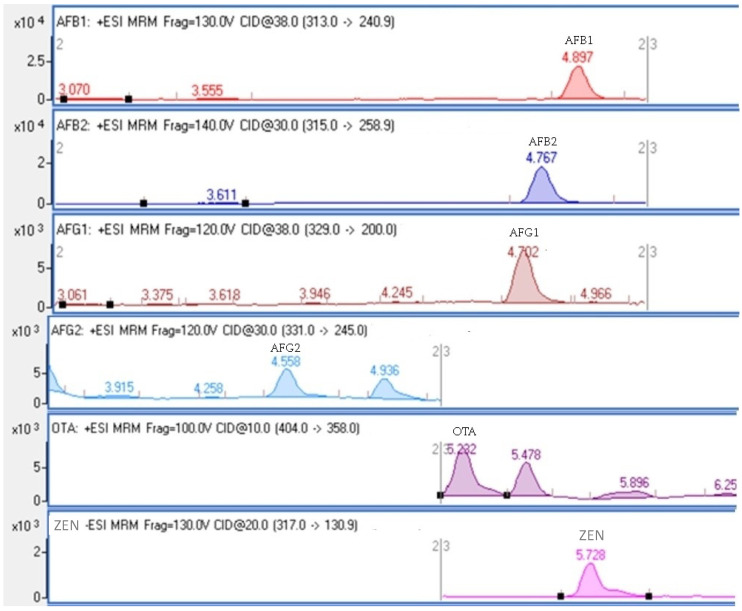
LC-MS/MS chromatograms of mycotoxins (AFB1, AFB2, AFG1, AFG2, OTA, and ZEN).

**Table 1 microorganisms-11-01786-t001:** Validation parameters of the mycotoxins.

	Linear Range (µg/L)	R^2^	RSD (%)	% Rec	LOD (µg/kg)	LOQ (µg/kg)
AFB1	0.05–1	0.998	4.95	82	0.021	0.071
AFB2	0.05–1	0.997	3.85	83	0.036	0.122
AFG1	0.05–1	0.998	4.83	90	0.038	0.126
AFG2	0.05–1	0.999	5.53	88	0.036	0.121
OTA	0.05–1	0.999	4.3	75	0.043	0.145
ZEN	0.5–10	0.999	2.78	108	0.403	1.343

**Table 2 microorganisms-11-01786-t002:** Mycotoxin mean ± SE (µg/kg) values of spice samples.

Spice	Brands	AFB1 (µg/kg)	AFB2 (µg/kg)	AFG1 (µg/kg)	AFG2 (µg/kg)	OTA (µg/kg)	ZEN (µg/kg)
Black pepper	A	0.83	0.68 ± 0.14	ND	1.04 ± 0.30	ND	ND
	B	3.93 ± 1.92 ^a^	0.16	ND	ND	7.89 ± 0.97	ND
	C	0.20 ± 0.05 ^b^	ND	0.63 ± 0.29	1.52 ± 0.80	ND	ND
	D	ND	ND	ND	0.14	5.46 ± 2.83	ND
	Total	2.48 ± 1.22	0.61 ± 0.14	0.63 ± 0.29	1.17 ± 0.36	6.27 ± 1.89	ND
Red pepper	A	0.87 ± 0.17 ^b^	0.28 ± 0.02	0.15 ± 0.01 ^b^	ND	ND	ND
	B	6.01 ± 2.99	0.55 ± 0.13	0.30 ± 0.04 ^a^	1.11 ± 0.19	11.68 ± 4.42	ND
	C	6.03 ± 4.18	0.75 ± 0.34	0.29 ± 0.03 ^a^	0.57	5.70 ± 1.94	ND
	D	2.29 ± 0.72 ^a^	0.34 ± 0.06	0.19	0.34	11.51 ± 6.58	ND
	Total	3.76 ± 1.27	0.47 ± 0.08	0.26 ± 0.02	0.92 ± 0.18	10.82 ± 3.11	ND
Cumin	A	ND	ND	ND	ND	1.45 ± 0.04	ND
	B	3.24 ± 1.19	ND	ND	ND	4.41 ± 0.99	ND
	C	ND	ND	ND	ND	ND	ND
	D	ND	ND	ND	ND	1.51 ± 0.47	ND
	Total	3.24 ± 1.19	ND	ND	ND	3.19 ± 0.72	ND
Turmeric	A	ND	ND	ND	0.54 ± 0.19	0.49	6.20 ± 1.42
	B	3.42 ± 2.72	ND	ND	0.22 ± 0.08	ND	ND
	Total	3.42 ± 2.72	ND	ND	0.43 ± 0.14	0.49	6.20 ± 1.42
Total		3.39 ± 0.88	0.50 ± 0.07	0.29 ± 0.03	0.92 ± 0.19	6.96 ± 1.45	6.20 ± 1.42

Different letters in the brands of spice samples in each column are statistically significant (*p* < 0.05). AFB1: Aflatoxin B1, AFB2: Aflatoxin B2, AFG1: Aflatoxin G1, AFG2: Aflatoxin G2, OTA: Ochratoxin A, ZEN: Zearalenone, ND: Not detected.

**Table 3 microorganisms-11-01786-t003:** The mean ± SE and minimum and maximum counts of *Staphylococcus* spp. and *Micrococcus* spp. of spice samples.

Spice	Brands	*Staphylococcus* spp.			*Micrococcus* spp.		
		Mean (log CFU/g)	Min (log CFU/g)	Max (log CFU/g)	Mean (log CFU/g)	Min (log CFU/g)	Max (log CFU/g)
Black pepper	A	<log 2	<log 2	<log 2	2.54 ± 0.06 ^c^	2.48	2.60
	B	4.01 ± 0.05 ^a^	3.60	4.37	4.27 ± 0.05 ^a^	3.90	4.75
	C	<log 2	<log 2	<log 2	<log 2	<log 2	<log 2
	D	2.96 ± 0.14 ^b^	2.30	3.94	3.47 ± 0.18 ^b^	2.00	4.53
Red pepper	A	<log 2	<log 2	<log 2	2.54 ± 0.06 ^d^	2.48	2.60
	B	3.87 ± 0.10 ^a^	3.00	4.38	3.95 ± 0.12 ^a^	2.85	4.48
	C	3.07 ± 0.10 ^c^	2.30	3.75	3.37 ± 0.09 ^bc^	2.78	3.95
	D	3.48 ± 0.09 ^b^	2.90	4.26	3.60 ± 0.10 ^abc^	2.00	4.06
Cumin	A	<log 2	<log 2	<log 2	2.62 ± 0.25 ^b^	2.00	3.75
	B	3.42 ± 0.20 ^a^	2.48	4.08	4.00 ± 0.06 ^a^	3.41	4.45
	C	2.72 ± 0.34 ^b^	2.00	3.56	2.87 ± 0.27 ^b^	2.00	4.33
	D	3.03 ± 0.13 ^ab^	2.00	3.70	3.21 ± 0.15 ^b^	2.00	4.06
Turmeric	A	3.92 ± 0.21	2.00	4.72	3.99 ± 0.21	2.30	4.86
	B	4.08 ± 0.12	3.60	4.48	4.49 ± 0.11	3.51	5.07

Different letters in the brands of spice samples in each column are statistically significant (*p* < 0.05).

**Table 4 microorganisms-11-01786-t004:** Estimated daily intake (±SE) of mycotoxins of spices.

	Red Pepper	Black Pepper	Cumin	Turmeric
	EDI ng/kg bw	EDI ng/kg bw	EDI ng/kg bw	EDI ng/kg bw
AFB1	0.51 ± 0.17	0.34 ± 0.17	0.44 ± 0.16	0.47 ± 0.37
AFB2	0.06 ± 0.01	0.08 ± 0.02	-	-
AFG1	0.04 ± 0.003	0.09 ± 0.04	-	-
AFG2	0.13 ± 0.02	0.16 ± 0.05	-	0.06 ± 0.02
OTA	1.47 ± 0.42	0.85 ± 0.26	0.43 ± 0.10	-
ZEN	-	-	-	0.84 ± 0.19

EDI: estimated daily intake, AFB1: Aflatoxin B1, AFB2: Aflatoxin B2, AFG1: Aflatoxin G1, AFG2: Aflatoxin G2, OTA: Ochratoxin A, ZEN: Zearalenone.

**Table 5 microorganisms-11-01786-t005:** Margin of exposure assessment of mycotoxin levels in spice samples.

	BMDL ng/kg bw	Red Pepper		Black Pepper		Cumin		Turmeric	
		MOE	EDI ng/kg bw	MOE	EDI ng/kg bw	MOE	EDI ng/kg bw	MOE	EDI ng/kg bw
AFB1	400	784.31	0.51	1176.47	0.34	909.09	0.44	851.06	0.47
AFs	400	1670.33	0.24	1896.95	0.21	909.09	0.44	2500.00	0.16
OTA	14,500	9863.95	1.47	17,058.82	0.85	33,720.93	0.43	-	-

BMDL: benchmark lower dose limit, MOE: margin of exposure, EDI: estimated daily intake, AFB1: Aflatoxin B1, AFs: Total Aflatoxin, OTA: Ochratoxin A.

**Table 6 microorganisms-11-01786-t006:** The studies conducted on mycotoxin analysis in spices.

Sample	Methods	Mycotoxins	Range	Reference
Spices	LC-MS/MS	AFB1, AFG1, and OTA	3–19 µg/kg, 10–11 µg/kg, and 4–20 µg/kg, respectively.	Motloung et al. [18]
Black pepper	UHPLC–MS/MS	OTA and AFB2	1.85 µg/kg and 0.358 µg/kg, respectively	Pantano et al. [34]
Spices	HPLC-FLD	OTA	0.11–38.46 ng/g	Pickova et al. [35]
Commercially available spices	UPLC-MS/MS	AFB1, Total AF, OTA, and ZEA	193.4 µg/kg, 168.1 µg/kg, 7.1 µg/kg, and 30.6 µg/kg, respectively	El Darra et al. [36]
Black pepper	Quantitative ELISA	OTA	1.15 to 1.91 μg/kg	Nguegwouo et al. [37]
Black pepper, red pepper, and turmeric	HPLC	OTA	3.31 ng/g, 5.66 ng/g, and 2.77 ng/g, respectively.	Jalili [38]
Red pepper	HPLC	AFB1 and AFB2	15.51 μg/kg and 1.17 μg/kg, respectively.	Barani et al. [39]
Black pepper		AFB1 and AFB2	1.09 μg/kg and 0.21 μg/kg, respectively	
Dried peppers	HPLC	AFB1, AFB2, AFG1, AFG2, and OTA	3.37, 0.45, 0.67, 0.073, and 7.15 ng/g, respectively.	Jalili and Jinap [40]
Paprika and chili pepper		AFB1, AFB2, AFG1, AFG2, OTA, and ZEA	1.2–64.4 μg/kg, <LOQ–1.2 μg/kg, <LOQ–2.5 μg/kg, <LOQ μg/kg, 4.3–118.7 μg/kg, and <LOQ–114.3 μg/kg, respectively.	Santos et al. [41]
Paprika and chili spices	HPLC	AFB1, AFB2, AFG1, AFG2, OTA, and ZEA	<LOQ–2.66 μg/kg, <LOQ–0.54 μg/kg, <LOQ–0.51 μg/kg, <LOQ–4.77 μg/kg, <LOQ–281 μg/kg, and <LOQ–131 μg/kg, respectively.	Santos et al. [42]
Chili, black, and white peppers	LC-MS/MS	AFB1 and AFB2	0.6 μg/kg and 0.6 μg/kg	Boonzaaijer et al. [43]
Paprika powders		OTA	2.1–8.0 μg/kg	
Red chili powders	HPLC-FD	AFB1, AFB2, AFG1, AFG2, and OTA	5.10 μg/kg, 0.32 μg/kg, 0.68 μg/kg <LOD, and 24.65 μg/kg, respectively.	Ozbey and Kabak [44]
Black peppers		AFB1, AFB2 and OTA	0.20 μg/kg, 0.05 μg/kg, and 1.82 μg/kg, respectively.	
Cumin		AFB1, AFB2 and OTA	0.58 μg/kg, 0.08 μg/kg, and 0.63 μg/kg, respectively.	
Pepper paste, pepper sauce, and red pepper flakes	ELISA	AFB1	<1.25 ppb to 4–5 ppb	Yentur et al. [14]
red pepper, black pepper and cumin		Total AF	6.36 µg/kg, 3.44 µg/kg, and 47.03 µg/kg, respectively.	Kurşun and Mutlu [45]
Red pepper	ELISA and HPLC	AFB1	ND (not detected) –32.2 µg/kg and ND–35.5 µg/kg, respectively.	Çolak et al. [46]
Black pepper			ND–9.5 µg/kg and ND–10.3 µg/kg, respectively.	
Black pepper, red pepper, cumin, turmeric	LC-MS/MS	AFB1	2.48, 3.76, 3.24 and 3.42, respectively	Present study
		AFB2	0.61, 0.47, ND and ND, respectively	
		AFG1	0.63, 0.26, ND and ND, respectively	
		AFG2	1.17, 0.92, ND and 0.43, respectively	
		OTA	6.27, 10.82, 3.19 and 0.49, respectively	
		ZEN	ND, ND, ND, 6.20, respectively	

## Data Availability

Data are provided in the manuscript.

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
