# Peer review of "Analysis of Multi-Mycotoxins in Commonly Consumed Spices Using the LC-MS/MS Method for Assessing Food Safety Risks"

_microorganisms, 2023, doi:10.3390/microorganisms11071786_

Round 1

Reviewer 1 Report

Manuscript ID: microorganisms-2481917

Title: " Analysis of Multi-Mycotoxins in Commonly Consumed Spices Using theLC-MS/MS Method for Assessing Food Safety Risks "

The authors present the results of mycotoxin analysis on one hundred and forty spice samples purchased in Ankara, using a LC MS/MS methodology..

The work is presented in a clear and organized way, although it has some repetition. One of the most evident cases is in the conclusions where the authors repeat entire sentences.

It also seems excessive the comparison that is made in the discussion with the results obtained by other authors. It would probably be better and the results easier to compare with the present study if the authors had put these results in a table.

There is some information that authors should add to the article. That is the case with the JSM 92 FO 9704 reagent, which should be explained what it consists of, and the authors should specify how the mobile phases used are constituted. The gradient is mentioned, but not the composition of the eluent.

In line 123 it is stated that validation parameters were obtained.  Method validation is a very important evaluation for an analytical article such as this. The authors have not done, at least do not present, a full validation. They should present the linearity data; in the recovery test they should present which concentration or concentrations were studied. In line 124 they do not specify to which parameter the relative standard deviation (RSD) refers.

Although the habits of the Turkish people may be very different from other peoples, the daily consumption of 10g of spices (line 157) per day per person, seems to me to be very high. It can be understood as the worst case.

It is very clear and worrisome the difference in the results obtained with brand B compared to what was obtained with the other brands. There must be a huge problem somewhere in the production process. The authors should have the responsibility to communicate the analyzed situation to the producers. This is also our role as researchers.

In my opinion, after the proposed corrections, the manuscript will improve its quality and could probably be accepted for publication in Microorganisms.

Author Response

Dear Reviewer, thank you very much for your valuable comments. The corrections have been made and specified in the article, and the responses to your comments have been uploaded to the system as a file.

Reviewer 2 Report

As a result of the review of the paper entitled "Analysis of Multi-Mycotoxins in Commonly Consumed Spices Using the LC-MS/MS Method for Assessing Food Safety Risks", a number of more or less detailed comments are presented below: 

Line (L) 12 - Due to the chemical structure and the accepted way of writing, the abbreviation ZEN type should be used in the case of zearalenone. I would suggest that such corrections be made throughout the text;

L 41 - In my opinion, the term mold fungi should be used, not the term mold itself, much less mold mycotoxins. I would therefore propose to improve the naming throughout;

L 54 - I miss one or two sentences about zearalenone prevalence. What is its adverse impact on the health quality of food products;

L 68 - What about ZEN?

L 70 - It should be decided and use the full name of mycotoxins or their abbreviations in the whole work, not both at the same time;

L 71 - I lack a justification for why a more thorough microbiological analysis has not been carried out. Only two genera were considered (why these ones?), and very dangerous species within these genera were not included. Such results would increase the scientific value of the work. For example, in the genus staphylococcus, 30% of species physiologically live on the skin, and only a few are very dangerous. Therefore, the reliability of the analyzed Staphylpcoccus spp./Micrococcus spp ratio is not very sensitive. After all, it is about the health quality of certain foodstuffs;

L 79 - Were the spices obtained from plant materials harvested in 2020 or 2021?

L 133 - I would suggest changing the subsection title to Bacteriological tests or Microbiota tests. Why, because no analysis of bacteria as such was performed, only microbiological assessment (of the type of bacteria) was performed.

LL 299, 309, 315, 339, 420 - The discussion chapter should not refer to the results obtained in Turkey, to the results obtained in the Czech Republic, Lebanon, South Africa, Pakistan or. These are completely different climatic-geographical conditions, as a result of which the prevalence of mycotoxicoses and bacteremia is very different. I think it would be better to refer only to the authors without indicating the place where the experiment was performed. Also, no attempt has been made to assess whether there is an interaction between the fungal kingdom and the bacterial kingdom. As a result, the "Discussion" chapter is very shallow and should be reworded;

L 393 - What does the mold fungus do here?

Author Response

(The authors gave the same response as above.)

Round 2

Reviewer 2 Report

OK